# AN AFROCENTRIC PERSPECTIVE ON ALGORITHM WATERMARKING OF AI GENERATED CONTENT

## ABSTRACT

Digital-driven misinformation, counterfeiting, and copyright violations have become a growing concern in Africa. The prevalence of Artificial intelligence content (AIGC) has the potential to widen its impact and create more challenges for the people on the continent. AIGC poses a dual challenge. First, creatives who have worked so hard to create a masterpiece see their work being illegally duplicated or used without their consent. The other unsuspecting individuals have fallen prey to misinformation caused by AIGC. The reason, amongst many, could be the regulatory gaps in the law governing data protection, copyright and even artificial intelligence. This paper argues that curating technical watermarking methodologies/techniques is insufficient, considering the uniqueness of the African continent. It further addresses the regulatory gaps by examining the existing laws and proposing an Afrocentric perspective on AIGC using Nigeria, Kenya, Egypt and South Africa as case studies.

## 1 INTRODUCTION

The technique of watermarking historically was first implemented in the 13th century by an Italian paper manufacturing company acc (2023). Unique designs or symbols were impressed into the paper during the production process Har (2021). The practice was unique in identifying quality and origin and deterring malicious individuals from counterfeiting the product Liu et al. (2025).The technique has evolved from a simple marker for determining paper authenticity to sophisticated digital security features Acc (2023). Watermarking can be engraved in varying visibility, from overt markers to opaque, hidden signals Sco (2024). For instance, Stock images might have a clear, opaque stamp that denotes their origin. In contrast, others might be integrated into their pixel structure or embedded within patterns of text punctuation, making them invincible to a casual observer.

Generative artificial intelligence (GenAI) such as DALL-E OpenAI (2025b), Midjourney MidJourney (2025) and ChatGPT OpenAI (2025a) can produce highly realistic content in varying formats such as text, image, video and audio Jiang et al. (2024). The same capacity allowing swift content generation can be exploited to create harmful or misleading information. AI-generated content (AIGC) has become increasingly prominent and has raised legal and ethical concerns on the African continentLi et al. (2024). As a legal concern, the ability of GenAI to mimic patterns so convincingly raises the issue of intellectual property. Firstly, individuals might falsely claim copyright over content generated by an AI system, Secondly, the original authors of the output from which the data was trained can identify their work. As an ethical issue, it could be weaponised to support disinformation or propaganda campaigns. The market for generative AI was predicted to increase to 50 billion by 2028MarketsandMarkets (2023) Which shows the increasing growth of the GenAI industry.

In summary, this work contributes the following to the existing body of knowledge:

1. To the best of our knowledge, it is the first Afrocentric-focused work on algorithm watermarking. We perceive watermarking from two perspectives: first, the attribution of Indigenous data that originates from Africa, and second, the methodology of AIGC verification..

2. We explore the regulatory gaps in data protection, intellectual property, and human rights protection using Nigeria, Kenya, South Africa, and Egypt as case studies to consider the rationale behind the prevalence of unattributed and harmful AIGC. We further map out best practices by recommending ways forward for the continent.

## 2 RELATED WORKS

Various scholars and organisations have designed various literature on algorithm watermarking methodologies. These methodologies have focused on techniques for ensuring robust watermarking metrics.

Zhengyuan Jiang (2024) In their work, they studied watermark-based, user-level attribution of AIGC. In their framework, users are issued a unique watermark (a bistring) stored in a centralised database when registering for a GenAI service. Every AIGC generated by the user carries a generalised watermark. The challenge with the work is that most GenAI companies allow for the anonymous usage of their tool, which might not be practicable in the real world. Kirchenbauer et al. (2024) built on the works of Atallah et al. (2001) And Chiang et al. (2004). Kirchenbauer et al. (2024). proposed a watermarking technique that influences token selection in text generation. They introduced a detection algorithm that identifies watermarked text without access to the model's internal parameters or its API. The framework was further tested without large language models such as stable diffusion, Midjourney, and DALL-E.

## 3 CHALLENGES AND LIMITATION

An adversary can easily erase the embedded watermark from the generated content and then use it freely without the service provider's regulation. The adversary can create illegal content with forged watermarks from another user, causing the service provider to make wrong attributionsLi et al. (2023). These works extensively address the technical framework for watermarking AIGC with limitations in considering the user's privacy, intellectual properties, or the possibilities of watermark mutation by users.

## 4 REGULATORY LANDSCAPE ON WATERMARKING

We developed four metrics to properly evaluate the regulatory framework governing the watermarking of AIGC in Africa. The aim of creating these metrics is to capture multiple facets of the current landscape. We admit that Africa's regulatory landscape regarding watermarking or algorithm watermarking/attribution matters may be outdated. This is because the laws were enacted long before the modern-day growth of artificial intelligence (AIGC) on content. As a result, there would be noticeable gaps in the frameworks. The metrics include:

1. Provision of watermarks, if any
2. Provision for AIGC
3. Institutional oversight on copyright materials
4. Judicial Opinion/position on algorithm watermarks, if any

### 4.1 NIGERIA

The Copyright Act 2022 has been revolutionary for protecting the rights of authors and creators in Nigeria Wysebridge Patent Bar Review (2025). However, under the Act, intellectual property safety provides no direct protection for using copyrighted material by unauthorised third parties. Rapid advancements in generative technologies characterised by replicating creators' works necessitate a broad interpretation of existing laws protecting digital watermarking to address the lacuna that could accrue as damages for creators. Section 5 of the Copyright Act provides that only legal persons (natural persons or corporate entities) are eligible for the production of copyrighted material. By its nature, AI-generated content borrows from the original works of others, making it difficult for a creator to rest under the scope of moral rights attributed to creators by section 14 of the Act. Section 14(b) further grants rights to original authors to seek legal redress for any modification of their original content. In defining 'copyright infringement', however, the Act does not make provision for machine reproduction of original content (s. 36), providing ample grounds for a legal argument that AI-generated content can thus be watermarked and passed off as the original work of its generator. The lack of originality in owning and training data on which generative models are run provides a basis for the argument that a derivative product (such as AI-generated content) may

be viewed as an inspiration and not an authentic concept. The core principle behind copyrighting works is incentivising people to create; such authorship/creatorship should be given a green light. This will serve as a nod to developers, researchers and individuals involved in the AI development and creation pipeline. However, the recommendations are that such legal protections be limited to a few years (5-7 years) [23].Amatika-Omondi (2025)

## 4.2 KENYA

Kenya's Copyright Act provides that the owner of an original work is a legal person or entity. The Act provides that a person for whom arrangements are provided for the production of a creative work may serve as its creator. In making a case for AI-generated content, creators who provide any necessary arrangement for the production of an expressive output, e.g., a media program, may be permitted to retain copyright over said works .Laws of Kenya (2001) Therefore, if the nature of the AI machine does not substantially derive from the works of others, i.e. a machine that a substantial amount of its output rests primarily on itself, then such generated work may be interpreted as not amounting to infringement and, thus, capable of copyright protections by its prompt-user. The courts and relevant authorities are likely to consider the extent/degree of human input, the novelness or originality of the said work and the purpose for its reproduction, e.g. commercially generated content is more likely to be favoured in the context of copyrighting protections.Kwang'a (2025)

While there are no clear-cut regulations against copyrighting, various interpretations of what constitutes original works in different jurisdictions could influence AI-generated content in the coming years. Institutions like the Kenya Copyright Board WKA Advocates (2025) are already examining the eligibility of AI-generated content for copyrighting and adjusting existing regulations that may redefine intellectual production or categorise algorithmic output as a distinct category.

## 4.3 EGYPT

The establishment of the Egyptian Intellectual Property Law of 2002 did not envision or make provisions for AI-generated content in the context of innovative/creative works. Article 4 of the Act provides that natural persons or legal entities are the two categories of persons that may apply for patents.Law on the Protection of Intellectual Property Rights (2002) Article 10 of the Act offers some protection for authors of original works to prevent the "... using, selling or distributing..." of a person's work by a third party without their authorisation. Law on the Protection of Intellectual Property Rights (2002) The judicial institution has yet to address intellectual ownership in the context of AI-generated material and, therefore, has not established a precedent regime for algorithmic watermarking on generative works. Recent efforts have been aimed at drafting amendments to these provisions, looking to Saudi Arabia's National Strategy of Data and AI to develop its IP rights and techniques, such as open licensing. Some schools of thought propose that the doctrine of first sale allows those possessing copyrighted material to exploit such for commercial purposes, although very little is said about actual ownership of derivative works. Still, fair use criteria may be examined in assessing the extent of originality and 'human participation' in the innovation process.Khalaf (2024) A typical example of this would be distinguishing the process of 'prompting', which can produce several different, randomised results from artistic reproductions, e.g. text, paintings, music, and designs that by their nature require a personalised element of style, tone, thematic elements and then 'reference' that is expressive in its most fundamental state. Illuminating the spirit of the copyright concept requires the undisputed existence of those distinguishing features that act as a human imprint on art.

## 4.4 SOUTH AFRICA

As is obtainable in other jurisdictions, South African laws do not recognise the copyrighted ownership of materials generated by artificial intelligence. The National Copyright Act,Republic of South Africa (1978) which applies to private and public entities, provides that a work must have origins reduced to its material form. and authored by one Indigenous to domiciled in South Africa or a member of the Berne Convention.Companies and Intellectual Property Commission (CIPC) (2025) Interestingly, the Act recognises that a work can be computer generated and makes provision to grant authorship to the person for whom the arrangements were made, the premise of which algorithmic watermarks can rely on in granting ownership. This implies the Act places greater pri-

ority on the economic theory as justifiable grounds for copyrighting, positioning creators behind algorithms as incentive contributors to its own work rather than other theories of appropriation and labour.Anonymous (2024) In future contexts, this categorisation makes a case for a loose interpretation that recognises the identification of an author of AI-generated work as the inventor or creator of such material. Where such information has been deliberately publicised by its owner or is made available in public records (in the context of personal information collection), the law permits the use of such data.Republic of South Africa (1978) Under the 'Terms of Use' provisions of OpenAI, input and output used to generate content is assigned to its generator subject to the condition that relevant and applicable country laws allow it.OpenAI (2025c)

## 5 ANALYSIS OF FINDINGS

Algorithm watermarks can be essential for helping users make informed choices about how they interact with AIGCs. However, these tools can violate users' rights and potentially process their personal data when maliciously weaponised or used as bait for unsuspecting users. In the four jurisdictions above, the processing of personal data requires consent. There is no provision for AIGC, regulations on algorithm watermarking, or decided cases on algorithm watermarking. This can be challenging because copyright laws are territorial in nature.Buick (2024) Most of these tools are built somewhere in the Global North, and None of the Centers are located in Africa,The Economist (2023) making it more difficult for institutional oversights to enforce compulsory localised watermarking.

As a result, if unauthorised reproductions of copyrighted material were to be carried out entirely in a continent whose law permits such use without rightsholder permission, there would be no copyright infringement in either country or within the territory.Peukert (2024) This is why the discussion on algorithm watermarking and trust in AIGC is more weighed on ethics than law: many companies will exploit regulatory gaps for their benefit.

Unlike in jurisdictions like China and the US, offices presiding over copyright matters have established legal requirements for publishing generated content to allow users to distinguish their nature and track authenticity. The OECD recommends that organisations using watermarking techniques subject generative models to assessments and media literacy to inform their audience. European Parliament (2023) The US seems to have taken a stance against granting authorship of AI-generated works with cases like Feist Publications v Rural Telephone Service Company.Feist Publications Inc. v. Rural Telephone Service Co. (1991) where the Court specified that copyright law only protects "the fruits of intellectual labour" that " are founded in the creative powers of the mind." " IP Think Tank (2025) Likewise, in Thaler v. Perlmutter, the US courts reestablished their position on an AI-generated painting, stipulating that it does not fulfil the conditions for "human authorship integral to copyrighted registration.Stephen Thaler v. Shira Perlmutter (2023) Similar provisions have been enforced in Australia's Acohs Pty Ltd v Ucorp Pty LtdAcohs Pty Ltd v Ucorp Pty Ltd (2012), where the Court pronounced that a work generated through computer "intervention" did not come under the legal protections reserved for human ingenuity. The Court looked to the level of activity demonstrated to ascertain the degree of authorisation rather than determining authorship based on the existence or absence of factors considered to prevent infringement. In contrast to the United States, the European Union has adopted a legislative approach, with the passage of the EU Artificial Intelligence Act in March and establishing an AI Office to enforce it. China's government has already introduced mandatory watermarking, and California wants to do the same.

It is also observed that all available resources about AIGC algorithm watermarking have been one-sided. That is companies providing watermark metrics and tools to allow attribution and content source for their LLM give no room for disclosure of the source.Jernite (2023) For instance, OpenAI in the paper introducing GPT-4 revealed only that the data on which the model had been trained was a mixture of "publicly available data (such as internet data) and data licensed from third-party providers"Byrd (2023). Some AI company justifies their decision to be secretive regarding details of their training data based on concerns regarding "the competitive landscape and the safety implications of large-scale models", with no further explanation.OpenAI (2023) They argue that sharing further details regarding their training data would facilitate replicating their cutting-edge AI models while releasing detailed information would enable careless or malicious actors to develop their powerful AI models more efficiently.The Verge (2023)

Watermarking should be double-sided. One is attributing the content of AI-generated works, and the second is attributing source data, especially those that have been copyrighted and used in the jurisdiction of their origin. Although copyrights are only limited to the country where the copyright is registered, the duty is on the company to ensure that copyrighted materials are not at all or not attributed as their own when watermarking.

## 6 CONCLUSION

Our work views watermarking to serve two dual but broad purposes. First, as a means and secondly, as an end. As a means, algorithm watermarking serves as a robust method for verifying the authenticity of AIGC ensuring that outputs (text, video, image or audio), can be traced back to their source. As an end, to recognize and attribute the contributions of the training data sets of the original/indigenous owners of that data instrumental in the model training. With this, it helps to establish a transparent and unbroken chain of provenance. With this, it validates the legitimacy of the content and simultaneously preserving the historical context of the data used. This dual functionality promotes accountability and trust, offering a clear pathway to resolve disputes related to data usage and copyright claims, and ultimately fostering a more ethical and transparent landscape in the realm of AI development and deployment. Africa is a unique continent and the approach toward the watermarking of AIGC must take a unique approach. Considering the level of digital literacy in the continent is an indication that there will be a rise in risk fostered by AIGC. This would be an issue both for creatives both as the owners of data and the users of AIGC.

## 7 RECOMMENDATION

1. Governments of African countries should fund the development of open-source AI Afro-centric watermarking tools and oversee the compensation of artists for royalties when copyrighted or indigenous data is used to train algorithms.

2. Copyright regulations should contain such requirements, like data protection regulations that support/restrict cross-border data flow. For example, a provision on cross-border mobility of personal data should be included in copyright regulations, allowing stronger cross-sector/cross-border collaborations.

3. We recommend establishing and collaborating with African countries' copyright repositories/Indigenous databases. That way, it is easy to attribute copyrighted materials and trace their source when generated by AIGC. AI companies should prioritise collaborating with copyright, trademark, and digital commons repos in Africa.

4. Guidelines on identifying watermarks on AIGC and alternated watermarks.

5. Investment in AI literacy is essential to decipher between original content and AIGC. Multi-stakeholder collaboration is encouraged to achieve this.

6. Variation of models deployed to the African market.

7. Regulatory frameworks could demand/require companies to track and list significant datasets used in AI training. We recommend enforcing AI watermarking standards.

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
