# OpenReview forum: "An Afrocentric Perspective on Algorithm Watermarking of AI-generated Content"
_ICLR.cc/2026/Conference — Submitted to ICLR 2026_

### Official Review · Reviewer_UsNU · 2025-10-21

**Soundness:** 1
**Presentation:** 2
**Contribution:** 2
**Rating:** 2
**Confidence:** 3

**Summary:**

This paper explores an important and timely topic: examining AIGC algorithm watermarks from an Afrocentric perspective. The core argument of the paper is that purely technical watermarking solutions are insufficient in the unique context of the African continent, and regulatory loopholes and legal frameworks must be taken into account. The paper supports this argument through case studies of Nigeria, Kenya, Egypt, and South Africa, and offers policy recommendations.

The main strength of the paper lies in its novel perspective and the significance of the issue. Shifting the focus of the watermark discussion from a purely technical level to geopolitical, legal, and ethical dimensions, particularly focusing on the specific challenges of the Global South and Africa, is a very valuable contribution. The paper is well-structured, and both the introduction and conclusion sections effectively explain the research motivation and contributions.

However, the paper has some fundamental flaws that make it currently not meet the acceptance standards of ICLR. The most significant drawback is the lack of close connection with the machine learning community and substantial technical contributions. The paper reads more like a legal policy analysis article than a machine learning research conference paper. Additionally, the paper is lacking in rigor, including a weak literature review, insufficient depth in case studies, and a disorganized reference list.

**Strengths:**

1. Novelty & Significance:
The paper's topic is novel. It is the first (to the author's knowledge) systematic study from an African perspective on AIGC watermarks. It successfully elevates the discussion beyond technology to touch upon key issues such as data ownership, intellectual property, and regulation, which are crucial for establishing a truly global and ethical AI ecosystem.
2. Clarity & Readability
The paper is generally well-written and the language is smooth.

**Weaknesses:**

1. Deviates from the conference's mainstream interests
Despite its novel perspective, the paper's direct contribution to the field of machine learning is limited. ICLR readers expect to see algorithms, theoretical analysis, experimental validation, or in-depth technical discussions. This paper mainly conducts legal text analysis and policy recommendations, which deviates from the conference's mainstream interests.
2. Technical Quality & Correctness
The paper is not a technical one, so it is difficult to evaluate it from the perspectives of algorithm correctness or experimental design.
However, the rigor of its argumentation process is questionable. The four evaluation metrics proposed in Part 4 are reasonable, but the subsequent country-by-country analysis appears superficial, merely reiterating existing legal provisions without in-depth critical analysis or novel insights that directly link these laws to the challenges of watermarking technology. For instance, the paper points out the conflict between the "natural person" requirement in the law and AIGC, but this is a general issue and does not delve into its specific manifestations or solutions in the African context.
3. Clarity & Readability
The paper is generally well-written and the language is smooth.
However, there are serious problems with the reference format. The in-text citations (e.g., acc(2023), Sco(2024)) do not correspond to the reference list at the end of the paper, and the list format is extremely inconsistent (containing a large number of garbled characters and broken URLs), which seriously affects the academic rigor and readability of the paper.
4. Related Work
The literature review in the paper is very weak. It cites a few non-academic sources (such as blog posts, museum websites) on the history and technical overview of watermarks, but there is a severe lack of citations to important academic literature in the ICLR-related field (e.g., adversarial attacks, robustness studies, verifiable claims on AIGC watermarks, etc.). This makes the paper disconnected from the existing discourse in the machine learning community.

**Questions:**

1. Clarify the paper's positioning and target readers: The paper should clearly state that it is a legal policy paper. During the revision, the depth of policy analysis should be significantly enhanced instead of attempting to cater to technical conferences.
2. Deepen the analysis rather than merely describe: The case studies of the four countries need to go beyond the description of legal provisions. It is necessary to delve into:
- How are these legal loopholes exploited by specific AIGC application cases?
- How do the legal differences among different countries affect the deployment of cross-border AIGC services?
- How does the specific implementation of watermarking technology (such as the "double-sided watermark" mentioned in the text) interact with local laws? What specific obstacles will be encountered?
3. Standardize the reference format: References must be organized strictly in accordance with academic norms, ensuring that the citations in the text correspond one-to-one with the list at the end of the paper, and that the format is uniform and the information complete.
4. Strengthen the feasibility of arguments and suggestions: The policy recommendations (Part 7) are a good starting point, but they can be more specific.

---

### Official Review · Reviewer_ruhE · 2025-10-28

**Soundness:** 2
**Presentation:** 1
**Contribution:** 1
**Rating:** 0
**Confidence:** 4

**Summary:**

This paper provides the first Afrocentric-focused work on algorithm watermarking of AI-Generated Content (AIGC) , examining the regulatory gaps in data protection, intellectual property, and human rights in African countries like Nigeria, Kenya, Egypt, and South Africa. It proposes a dual purpose for watermarking: as a means for verifying AIGC authenticity and as an end for attributing indigenous data used in model training, ensuring a transparent chain of provenance. The work concludes with recommendations for African governments to fund Afrocentric watermarking tools, establish collaboration with copyright repositories, and enforce AI literacy and watermarking standards to address the unique challenges of the continent.

**Strengths:**

1. This paper is easy to follow.
2. The AIGC watermarking problem in Africa is an important problem.
3. This paper consider different areas in Africa.

**Weaknesses:**

1. The technical contributions of this paper are very limited.
2. There are obvious typos in the reference.
3. The paper is short and the content does not support it to appear on the top conference like ICLR.

**Questions:**

1. What are the technical details of the watermarking methods mentioned in this paper?

2. For related work part, there are more works that are not mentioned?

---

### Official Review · Reviewer_BKz3 · 2025-10-31

**Soundness:** 2
**Presentation:** 1
**Contribution:** 1
**Rating:** 2
**Confidence:** 4

**Summary:**

This paper presents an Afrocentric perspective on the watermarking of AI-generated content (AIGC), focusing on regulatory challenges and legal frameworks in Nigeria, Kenya, Egypt, and South Africa. The authors argue that technical watermarking solutions alone are insufficient for the African context and that existing intellectual property and data protection laws must be re-examined.

**Strengths:**

The paper offers an generally interesting Afrocentric perspective on AI-generated content attribution, providing a comparative analysis of African legal frameworks and their challenges. Its dual framing of watermarking shows strong ethical awareness, while recommendations for local tools, collaboration, and AI literacy add practical relevance. The inclusion of global comparisons offers useful context.

**Weaknesses:**

- Despite references to the concepts of watermarking, the paper lacks a detailed examination of algorithmic or mathematical watermarking techniques. Nowhere are the mechanics, robustness, or vulnerabilities of actual watermarking methods described or analytically evaluated.

- The paper is entirely qualitative. There are no experimental results, benchmarks, ablation studies, or even simple empirical case analyses illustrating how watermarking (or its absence) affects copyright enforcement or content verification in Africa.

- There are no figures, tables, or diagrams at all, not even a comparative table of legal frameworks across the examined countries, nor a schematic of the proposed Afrocentric watermarking eco-system. I mean these visual aid is very important for this type of contribution.

- The related work section reviews only a few methods (I recommend to carefully do an addon survey), relying on high-level summaries rather than critical dissection of algorithmic advances or weaknesses.

- There is repetitive narrative, especially in the problem framing (Sections 1, 5, and 6). Several paragraphs reiterate the same point about the limits of Western legal frameworks for AI in Africa without bringing in new technical or analytic insight.

The paper omits any mention of how watermarking techniques in AI-generated content might be benchmarked, i.e., what accuracy, robustness measures, or adversarial removal scenarios are relevant or could be evaluated in the African context, especially since these are core issues in the field.

**Questions:**

To what extent do the authors believe technical advances in watermarking (e.g., robust invisible watermarking, adversarial removal resistance) could be adapted or specifically tailored for African data regimes and legal systems? Do they foresee any unique technical hurdles or opportunities, such as for languages or data sources underrepresented in existing watermarking literature?

---

### Official Review · Reviewer_Emda · 2025-10-31

**Soundness:** 1
**Presentation:** 1
**Contribution:** 1
**Rating:** 0
**Confidence:** 5

**Summary:**

The paper presents a legal and policy-oriented discussion on the application of algorithmic watermarking for AIGC within the African context. The authors argue that purely technical solutions for watermarking are insufficient for the continent's unique challenges. The paper's main contribution is a review of the existing copyright and intellectual property laws in four African nations. It finds these legal frameworks to be outdated and ill-equipped to handle the complexities of AIGC.

**Strengths:**

The paper's primary strength is that it directs attention to the intersection of AIGC, watermarking, and intellectual property law within the African continent, a region often overlooked in global AI governance discussions. It correctly identifies that existing legal frameworks in the selected countries are not prepared for the challenges posed by generative AI.

**Weaknesses:**

1. The paper is a legal and policy analysis that contains zero technical substance. It does not engage with machine learning concepts, propose any computational methods, or present any empirical results. Submitting this work to ICLR demonstrates a profound misunderstanding of the conference's scope and audience.
2. The related works section is inadequate. It briefly mentions two technical papers without any depth and fails to engage with the vast and relevant literature on robust watermarking, AIGC detection, adversarial attacks against watermarks.
3. The core finding, that copyright laws written before the advent of modern generative AI are insufficient to govern it, is neither surprising nor novel. This is a well-established issue globally. Furthermore, the recommendations provided are highly generic and lack specific, actionable details derived from the preceding analysis.

**Questions:**

1. Could the authors please justify why they considered ICLR an appropriate venue for a paper that contains no technical contributions?
2. The analysis of the regulatory landscape concludes that existing laws are inadequate. Could the authors propose a specific, novel legal or technical-legal mechanism, informed by their Afrocentric perspective, that could address the identified gaps, rather than the list of general recommendations? For example, how should a watermarking algorithm be technically different to align with this perspective?

---

### Meta-Review · Area_Chair_AQy1 · 2025-12-30

**Summary:**

All reviewers considered the paper to be "not good enough".

Main concerns are:
- Lack of technical contributions
- Low quality of writing
- Inadequate coverage of related work
- Insufficient empirical results

**Reviewer Concerns:**

All, since there are no author responses.

**Reviewer Scores:**

N/A, since there are no author responses.

---

### Decision · Program_Chairs · 2026-01-26

Reject